# The Association Between the Hemodynamics in Anomalous Origins of Coronary Arteries and Atherosclerosis: A Preliminary Case Study Based on Computational Fluid Dynamics

**DOI:** 10.3390/bioengineering11121196

**Published:** 2024-11-26

**Authors:** Yuhao Wei, Haoyao Cao, Tinghui Zheng

**Affiliations:** 1Department of Mechanics & Engineering, College of Architecture & Environment, Sichuan University, Chengdu 610065, China; weiyuhaoo@stu.scu.edu.cn (Y.W.); hycao@stu.scu.edu.cn (H.C.); 2Yibin Institute of Industrial Technology, Sichuan University Yibin Park, Yibin 644000, China; 3Med-X Center for Informatics, Sichuan University, Chengdu 610041, China

**Keywords:** anomalous origin of coronary arteries (AOCA), atherosclerosis, hemodynamics, computational fluid dynamics

## Abstract

Patients with anomalous coronary artery origins (AOCA) exhibit a higher risk of atherosclerosis, where even minimal stenosis may lead to adverse cardiovascular events. However, the factors contributing to this heightened risk in AOCA patients remain unclear. This study aimed to investigate whether an AOCA patient is more prone to stenosis occurrence and its progression in view of hemodynamics. A patient whose left circumflex artery originated from the right coronary sinus with a mild stenosis in the left anterior descending (LAD) artery and a healthy individual were included in this study. Two additional models were developed by removing stenosis from the patient model and adding a corresponding stenosis to the healthy model. Additionally, the inlet flow waveforms for the left and right coronary arteries were swapped in both the patient and healthy models. Results indicated that the AOCA patient without stenosis demonstrated higher wall pressure (LAD: 95.57 ± 0.73 vs. 93.86 ± 0.50 mmHg; LCX: 94.97 ± 0.98 vs. 93.47 ± 0.56 mmHg; RCA: 96.23 ± 0.30 vs. 93.86 ± 0.46 mmHg) and TAWSS (LAD: 24.41 ± 19.53 vs. 13.82 ± 9.87 dyne/cm^2^, *p* < 0.0001; LCX: 27.21 ± 14.51 vs. 19.33 ± 8.78 dyne/cm^2^) compared to the healthy individual, with similar trends also observed in stenotic conditions. Significant changes in the LCX flow distribution were also noted under varying pulsatile conditions (LCX: 18.28% vs. 9.16%) compared to the healthy individual. The high-pressure, high-shear hemodynamic environment in AOCA patients predisposes them to atherosclerosis, and the unique geometry exacerbates hemodynamic abnormalities when stenosis occurs. Clinicians should closely monitor AOCA patients with stenosis to prevent adverse cardiovascular events.

## 1. Introduction

Anomalous origin of the coronary artery (AOCA) refers to the abnormal origin, course, or distribution of the coronary arteries and is classified as a congenital cardiovascular malformation with a low incidence rate [1,2,3]. While most AOCA cases are benign and do not produce clinical symptoms, studies indicate that individuals with anomalous coronary artery origins face a higher risk of developing atherosclerosis compared to the general population, particularly those with an anomalous origin of the left coronary artery from the right coronary sinus [4,5]. Once atherosclerosis develops in AOCA patients, even minor stenosis can lead to severe cardiovascular events, including myocardial ischemia and sudden cardiac death [5].

Current research on AOCA and coronary atherosclerosis is primarily clinical. Some researchers suggest that adverse events in AOCA patients with coronary stenosis are primarily due to abnormal vascular morphology, such as increased vessel curvature [6,7]. However, it is not entirely clear whether these morphological differences are the main contributors to the development of coronary atherosclerosis in AOCA patients, as normal coronary arteries can also exhibit high curvature and large bifurcation angles, which are known to promote atherosclerosis [8,9]. Another possible reason that AOCA patients are more susceptible to coronary atherosclerosis may be due to altered blood flow patterns. The abnormal course of the coronary artery may be influenced by pressure from the aorta, pulmonary artery, or other cardiac regions, inevitably affecting the blood flow waveform, which may contribute to plaque development.

In fact, there are inherent differences in blood flow waveforms between the left and right coronary arteries in individuals with normal coronary origins. Clinical observations reveal a lower incidence of right coronary artery stenosis, suggesting that the right coronary artery may have a more “stable” blood flow pattern [10,11,12]. Conversely, among AOCA patients, the left coronary artery originating from the right sinus is more prone to severe cardiovascular events compared to the right coronary artery originating from the left sinus. However, the precise factors leading to the increased risk of atherosclerosis in AOCA patients remain unclear.

It is well established that the hemodynamic environment within coronary arteries is closely linked to the development of atherosclerosis [13]. Therefore, this study selected a case of an anomalous left circumflex artery originating from the right sinus, with a course posterior to the aorta (traversing behind the aortic root to reach its normal anatomical position), exhibiting mild stenosis (degree of stenosis: 14%), and compared it to a case of a healthy individual. On this basis, we removed the stenosis in the patient model and added stenosis at the corresponding location in the healthy model. By comparing patient and healthy models both with and without stenosis, we aimed to investigate the impact of anomalous coronary artery morphology in the AOCA patient on coronary hemodynamics and plaque progression. Additionally, by setting different coronary flow waveforms, we explored the effect of coronary flow waveforms on the coronary arteries of the AOCA patient and healthy individual. Using computational fluid dynamics (CFD), we obtained the hemodynamic environment within the coronary arteries under these conditions. We hope to provide a theoretical basis for the diagnosis and treatment of patients with similar conditions.

## 2. Materials and Methods

### 2.1. Study Population

In this study, we selected a patient with an anomalous origin of the coronary artery for hemodynamic analysis. The patient’s left circumflex artery originated from the right coronary sinus and had a course posterior to the aorta, reaching the left coronary side. Initial examination revealed a non-obstructive stenosis of 14% in the left anterior descending artery (LAD), and the patient was re-hospitalized within five years due to coronary artery disease. Additionally, a healthy individual was included in the study as a control. The baseline characteristics are presented in Table 1.

The patient-specific computed tomography angiography (CTA) images and clinical data used in this study were sourced from the West China Hospital, Sichuan University. All participants adhered to the principles outlined in the Declaration of Helsinki, and the requirement for informed consent was waived due to the retrospective nature of the data collection. The study was approved by the Ethics Committee of West China Hospital, Sichuan University. Since all the data were collected from retrospective anonymized databanks and as our study was purely observational, patient approval and informed consent were waived.

### 2.2. Reconstruction of 3D Models

In this study, the patient’s computed tomography angiography (CTA) images and blood pressure from the initial consultation were utilized for subsequent model reconstruction and hemodynamic simulations. Based on the patient’s CTA images, a three-dimensional (3D) model of the coronary arteries, including the coronary artery tree and the aortic root, was reconstructed using the open-source software SimVascular 23.03.27 [14]. During the reconstruction, we retained only the proximal right coronary artery (RCA), left main coronary artery, proximal LAD, and proximal circumflex artery (LCX). The specific procedures for the reconstruction of the 3D models are illustrated in Figure 1. A more detailed description of the specific modeling process can be referred to in the studies by Cao et al. [15,16].

The meshes were generated using the open-source mesh generator TetGen. Based on previous grid independence studies and the SimVascular user guides, the unstructured tetrahedral mesh sizes were set to 0.04 mm for the coronary arteries and 0.2 mm for the aorta, with four boundary layers appended [15,16]. The initial height ratio of the boundary layer mesh was set to 0.5, and the decreasing ratio was set to 0.6. After meshing, the total number of elements was 1,553,994 for the patient model and 1,708,825 for the healthy control model. The grid independence analysis is presented in Table 2, which revealed that, between the analysis mesh and the 3-million-element mesh, the hemodynamic parameters, including the LAD flow rate and pressure drop at the inlet and outlet, differed by only 1.2% and 0.3%, respectively, indicating that the mesh density used in this study was sufficient for further analysis. It should be noted that the difference in the number of elements between the patient and healthy control models was primarily due to geometric shape variations rather than differences in mesh size.

### 2.3. Governing Equations

In this study, blood was assumed to be a continuous, incompressible, homogeneous Newtonian fluid, and the blood flow was assumed to be laminar. The governing equations are as follows:(1)$$\rho \frac{d\vec{v}}{dt}+\rho \left(\vec{v}\cdot \nabla \right)\vec{v}+\nabla p-\mu \Delta \vec{v}=0$$
(2)$$\nabla \cdot \vec{v}=0$$
where $\vec{v}$ represents the velocity vector of the fluid, *p* represents the pressure, *ρ* is the blood density (1.04 g/cm^3^), and *μ* is the dynamic viscosity of blood (0.04 dyne/cm^2^). In this study, a rigid wall assumption and no-slip boundary condition were assumed [17].

### 2.4. Boundary Conditions

At the inlet of the aorta, an ideal aortic physiological flow waveform was applied. At the aortic outlet, a Windkessel RCR model was applied to simulate the viscous resistance of the proximal downstream arteries (R_p_), the compliance of all vessels (C), and the resistance of the distal capillary and venous circulation (R_d_) [18]. Based on the study by Sankara et al., the ratio of proximal resistance R_p_ to distal resistance R_d_ was set to 0.09:0.91, and C was set to 0.001 cm^5^/dyne [18].

Notably, research by Ceserani et al. demonstrated that the LPN model achieves a strong correlation with clinical data when simulating coronary boundary conditions in AOCA patients [19]. Therefore, a lumped parameter network (LPN) model was applied to simulate the coronary artery outlet boundary conditions due to the lack of clinical ultrasound or 4D Flow data for this AOCA patient [20]. According to the study by Boger et al., the ideal coronary flow accounts for approximately 4% of the total cardiac output [21], so the total coronary resistance (*R_cor_*) could be estimated by
(3)$$R_{total}=R_{cor}+R_{aorta}=\frac{P_{mean}}{Q_{total}}$$
(4)$$R_{cor}:R_{aorta}=Q_{aorta}:Q_{cor}$$
where *P_mean_* represents the mean blood pressure, and *Q_cor_* represents the coronary flow rate. Using the patient’s actual blood pressure, we estimated total coronary resistance based on the aforementioned formula. After determining the total coronary resistance, we allocated the resistance at each coronary outlet in accordance with Murray’s law, proportional to the radius raised to the power of 2.6. In the LPN model, the resistance at each coronary outlet comprised coronary arterial resistance (R_a_), microcirculatory resistance (R_a-micro_), and coronary venous resistance (R_μ_), with coronary microcirculatory compliance (C_a_) and myocardial compliance (C_im_) also included. According to previous studies, the ratios for R_a_: R_a-micro_: R_μ_ were set at 0.35:0.50:0.15 and for C_a_: C_im_ at 0.11:0.89 [18,20]. These boundary conditions were applied across all computational models in this study. Additionally, a myocardial pressure function (P_im_) was applied at each coronary outlet to simulate the phase difference between coronary blood flow and aortic blood flow. Further details about the boundary condition setup can be found in the SimVascular user guides and the studies by Cao et al. and Chen et al. [15,16,22,23].

According to the SimVascular user guides and previous studies [15,16,22], each cardiac cycle was divided into 500 timesteps with a time step size of 0.002 s, and 12 cardiac cycles were simulated in total. The results from the 12th cardiac cycle were used for further analysis.

### 2.5. Study Design

#### 2.5.1. Effects of the Abnormal AOCA Morphology on the Coronary Hemodynamics

The AOCA patient in this study has a mild stenosis (14% degree of stenosis) in the LAD branch. We removed the plaque from the LAD branch of this AOCA patient (Patient no plaque) and compared the results with a healthy model to isolate the influence of abnormal coronary morphology on the coronary hemodynamic environment.

#### 2.5.2. Impacts of the Abnormal AOCA Morphology on the Progression of Coronary Plaque

To investigate how the anomalous coronary origin affects overall coronary hemodynamics and plaque progression, we developed an additional model (healthy with plaque) where the same stenosis was artificially introduced into the healthy model at the corresponding LAD location, approximately 1.2 cm downstream from the left coronary artery ostium, as shown in Figure 2. Consequently, this study thus included a total of four models for further analysis.

#### 2.5.3. Influences of the Coronary Waveforms’ Morphology on AOCA Patient

Under typical conditions, the left coronary artery displayed a single-peak flow waveform, while the right coronary artery exhibited a double-peak waveform (see Figure 3). To assess the effect of waveform variation on coronary hemodynamics, we swapped the flow waveforms between the left and right coronary arteries. Following the swap, the left coronary artery displayed a double-peak pattern, and the right coronary artery exhibited a single-peak characteristic. Hemodynamic simulations were then performed on both the patient model and healthy control, with the swapped waveforms shown in Figure 3.

In total, six cases were simulated in this study: the patient model (Patient), the patient model without stenosis (Patient no plaque), the healthy control model (Healthy control), the healthy control model with stenosis (Healthy with plaque) were analyzed under normal coronary flow waveforms, while the patient model with swapped coronary waveforms (Patient-adjusted), and the healthy control model with swapped coronary waveforms (Healthy-adjusted) were also simulated.

### 2.6. Hemodynamic Parameters

Time-averaged wall shear stress (TAWSS) refers to the average wall shear stress at various points on the vessel wall over one cardiac cycle, reflecting the distribution of wall shear stress throughout the cycle. The complex regulatory effects of wall shear stress (WSS) on atherosclerosis have been widely reported [13,24]. Habib et al. demonstrated that lower WSS promotes plaque progression, while higher WSS makes plaques more vulnerable and unstable [13]. Xiang et al. found that excessively low (<4 dyne/cm^2^) or high (>40 dyne/cm^2^) TAWSS can similarly lead to platelet aggregation or destructive remodeling of plaques [25]. *TAWSS* is defined as follows:(5)$$TAWSS=\frac{1}{T}\int_{0}^{T}\left|WSS\right|dt$$

Oscillatory shear index (OSI) refers to the ratio of wall shear stress at various points on the vessel wall over one cardiac cycle. Higher OSI typically indicates disturbed blood flow and is strongly associated with an increased risk of thrombosis [25]. OSI is defined as follows:(6)$$OSI=\frac{1}{2}\left(1-\frac{\left|\int_{0}^{T}WSSdt\right|}{\int_{0}^{T}\left|WSS\right|dt}\right)$$

Endothelial cell activation potential (ECAP) represents the possibility of thrombus formation, while a larger ECAP typically indicates a higher risk of thrombus formation [26]. ECAP was defined as:(7)$$ECAP=\frac{OSI}{TAWSS}$$

In addition, helicity was also assessed to evaluate the helical flow in the coronary arteries. Helicity was defined as:(8)$$Helicity=\vec{v}\cdot \left(\nabla \times \vec{v}\right)$$

### 2.7. Statistical Analysis

All data processing and visualization were conducted by ParaView 5.10 (Kitware. Inc., New York, NY, USA) and GraphPad Prism 10 (GraphPad Software, Boston, MA, USA).

## 3. Results

### 3.1. The Impact of Anomalous Coronary Artery Morphology on Coronary Hemodynamics

The hemodynamic distribution of the TAWSS, wall pressure, ECAP, and pressure drop between the abnormal and normal coronary arteries without stenosis are shown in Figure 4. Overall, although no difference was observed in ECAP, significant differences were still observed in the TAWSS, wall pressure, and pressure drop between the two models. Compared to the normal coronary artery, the abnormal coronary artery exhibited markedly higher spatially-averaged TAWSS in the LAD and LCX (LAD: 24.41 ± 19.53 vs. 13.82 ± 9.87 dyne/cm^2^; LCX: 27.21 ± 14.51 vs. 19.33 ± 8.78 dyne/cm^2^), while no substantial difference was noted in the right coronary artery (RCA, 11.93 ± 10.26 vs. 13.33 ± 6.48 dyne/cm^2^).

Furthermore, the abnormal coronary artery showed higher average wall pressure in the LAD (95.57 ± 0.73 vs. 93.86 ± 0.50 mmHg), LCX (94.97 ± 0.98 vs. 93.47 ± 0.56 mmHg), and RCA (96.23 ± 0.30 vs. 93.86 ± 0.46 mmHg) compared to the normal coronary artery. Most of the LAD, LCX, and nearly the entire RCA of the abnormal artery were consistently exposed to a higher-pressure environment, as shown in Figure 4B. Similarly, the pressure drops across the inlet and outlet followed this trend, with the abnormal arteries demonstrating greater pressure drops in the LAD (2.85 vs. 1.75 mmHg) and LCX (3.35 vs. 1.70 mmHg), where the abnormal LCX nearly doubled the pressure drop of the healthy LCX (Figure 4D). However, no obvious differences in pressure drop were observed between the abnormal and healthy RCA (1.38 vs. 1.51 mmHg).

### 3.2. Hemodynamic Changes in the Coronary Artery Morphology with Plaque

In the normal coronary artery, a region of low TAWSS was observed around the stenosis, as illustrated in Figure 5. In contrast, the stenosis in the abnormal coronary artery exhibited significantly higher TAWSS surrounding the stenosis. Overall, the abnormal coronary artery demonstrated markedly higher TAWSS than the normal coronary artery (LAD: 27.30 ± 17.65 vs. 13.35 ± 9.47 dyne/cm^2^; LCX: 31.43 ± 15.56 vs. 18.43 ± 8.15 dyne/cm^2^; RCA: 13.56 ± 10.18 vs. 12.92 ± 6.15 dyne/cm^2^), particularly in the LAD and LCX. The wall pressure distribution exhibited a similar trend, with higher values in the coronary arteries of the AOCA patient compared to the healthy individual (LAD: 94.60 ± 0.88 vs. 93.97 ± 0.60 mmHg; LCX: 93.92 ± 1.21 vs. 93.67 ± 0.51 mmHg; RCA: 95.28 ± 0.47 vs. 94.00 ± 0.52 mmHg). However, the distribution of OSI in the LCX and RCA showed no significant differences between the abnormal and normal models. Elevated OSI regions were only found in the LAD of the normal model, near the stenosis, while no abnormal OSI was observed in the abnormal model. The spatially averaged OSI across all branches in both models remained below 0.01. In addition, due to the overall low OSI levels, both the AOCA patient and healthy model maintained very low ECAP levels, with the spatially averaged ECAP values differing by less than 0.001 between the two models, showing no obvious difference.

The vortex-Q criterion distributions in the AOCA patient and healthy individual are presented in Figure 6. For the AOCA patient model, we consistently observed significantly more vortex structures and higher blood flow velocities in the LAD, LCX, and RCA during both acceleration and deceleration phases. In contrast, the healthy model showed substantially fewer vortices in the LCA during acceleration and a reduction in vortices in the RCA during deceleration. Additionally, compared to the healthy model and the AOCA patient model without plaque, the AOCA patient model with plaque exhibited a markedly increased helical flow in the LCX during the coronary flow acceleration phase, as shown in Figure 7.

### 3.3. Effects of Flow Waveform Morphology Changes on Coronary Perfusion in the AOCA Patient and Healthy Individual

The flow distribution and changes in the coronary branches before and after waveform adjustment in both the patient model and healthy control model are illustrated in Figure 8. It is evident that the flow distribution underwent substantial alterations following the waveform changes, as shown in Figure 8A. After waveform adjustment, the patient’s coronary flow exhibited a reduction in the LAD flow and an increase in the LCX flow (LAD: 37.52% vs. 32.10%; LCX: 26.52% vs. 31.36%). In contrast, the healthy control model showed the opposite trend, with an increase in the LAD flow and a decrease in the LCX flow (LAD: 27.36% vs. 30.49%; LCX: 33.74% vs. 30.65%). Figure 8B further depicts the normalized flow changes in the branches before and after waveform adjustment for both the patient and healthy models. Notably, the patient model showed greater changes across all branches compared to the healthy model, particularly in the LCX and right coronary artery (RCA) (LCX: 18.28% vs. 9.16%; RCA: 1.61% vs. 0.10%).

## 4. Discussion

Clinical evidence has suggested that the distinctive anatomical features and hemodynamic environment in some AOCA patients elevate their risk of adverse events after the formation of atherosclerotic plaques, and the morphology of coronary blood flow waveforms in these patients is more susceptible to changes [1,27]. Therefore, although the incidence of AOCA is relatively low, it is still of critical clinical importance to examine the hemodynamic effects of coronary morphology and changes in waveform morphology on atherosclerosis in AOCA patients to enhance the understanding. This study selected one case of an AOCA patient with mild stenosis in the LAD and one healthy individual to analyze the intraluminal hemodynamic changes under different flow waveforms and plaque conditions. The main findings of this study were as follows: compared to the healthy coronary artery, (i) in the absence of plaque, the AOCA patient exhibited greater pressure drops across coronary inlets and outlets, and significantly higher TAWSS in the left coronary artery; (ii) in the presence of stenosis, the TAWSS in the LAD and LCX of the AOCA patient model was also much higher than the stenosed normal model; and changes in blood flow waveform substantially impacted the flow distribution in the LAD and LCX of the AOCA patient.

Intraluminal pressure and TAWSS are two critical indicators closely associated with the onset and progression of atherosclerosis [28,29]. Prolonged high pressure can cause endothelial dysfunction, intimal thickening, inflammation, hardening, reduced elasticity, and structural damage to the arterial wall, ultimately leading to plaque formation [30,31], while TAWSS is considered as a significant factor influencing the progression of coronary atherosclerosis [13,24,32]. A greater pressure drop across the inlet and outlet suggests that the heart has to work harder to maintain physiological demands, increasing the myocardial load and placing additional stress on the vessels, which raises the risk of atherosclerosis and creates a vicious cycle [33], and elevated TAWSS can stimulate abnormal endothelial cell expression and platelet aggregation, thereby accelerating the atherosclerosis process [32]. The results of this study indicated that under the same boundary conditions, the pressure drops across all coronary branches in the AOCA patient without plaque were higher than that in normal coronary arteries, with the pressure drop in the anomalous LCX being twice as high as that in the healthy LCX. Additionally, large areas of high TAWSS in the LAD and LCX of the AOCA patient were also observed. This may suggest that the higher levels of TAWSS observed in AOCA patients are likely associated with the unique blood flow patterns resulting from their distinctive coronary morphology. That is to say, the coronary artery walls of the AOCA patient may be constantly exposed to high pressure and a high shear stress environment, and this adverse hemodynamic environment may possibly make AOCA patients with similar anatomies more susceptible to atherosclerosis compared to healthy individuals.

According to classical fluid mechanics, increased vortex structures within vessels can elevate energy loss in blood flow, thereby reducing flow efficiency. In this study, we observed that the AOCA patient exhibited more vortex structures throughout the cardiac cycle compared to the healthy individual, suggesting that the abnormal coronary morphology increased energy loss, thus imposing a higher workload on the heart to meet physiological demands. This increased workload may contribute to adverse cardiac remodeling and elevate coronary wall shear stress, potentially raising the risk of adverse events in AOCA patients.

Clinical findings suggest that compared to normal individuals, patients with AOCA may be more likely to develop atherosclerotic plaques at an earlier age. The study by Eid et al. observed that larger atherosclerotic plaques tend to form earlier in the anomalous vessels of AOCA patients, with over 80% of severe obstructive stenosis concentrated in the anomalous LCX [34]. Similarly, Samarendra et al. reported that, compared to other coronary arteries in the same patient, atherosclerotic stenosis appears earlier and is more severe in the anomalous arteries [35]. In addition, Mercado et al. also reported a case in which, despite normal lipid levels, a patient with AOCA exhibited extensive stenosis in the anomalous vessel [36]. Our study indicated that these phenomena may be associated with the suboptimal hemodynamic environment, which may lead to a higher risk of atherosclerosis development in AOCA patients, particularly those with this specific anomaly.

Under normal conditions, the differences in the waveforms of the left and right coronary arteries stem mainly from the varying pressures within the left and right ventricles. During systole, the left ventricular pressure is relatively high, resulting in more significant compression of the left coronary artery, which leads to a noticeable reduction in flow. In diastole, the pressure in the left ventricle drops considerably, restoring coronary perfusion, and thus, blood flow in the left coronary artery is primarily concentrated in diastole. The right coronary artery experiences less compression during systole because of the relatively lower pressure in the right ventricle, leading to a more uniform blood flow throughout the cardiac cycle. This results in a less pronounced decrease in flow during systole, in contrast to the left coronary artery [1]. However, for some AOCA patients, abnormal coronary artery anatomy may influence the way ventricular pressure impacts the coronary arteries, potentially altering the morphological characteristics of the coronary flow waveform [1,27]. Our findings revealed that the impact of flow waveform morphology on coronary perfusion cannot be overlooked, and the extent of this impact is related to the geometry of the vessel. When the flow waveform morphology changes, the LCX flow variation in the AOCA patient was nearly twice that of the healthy individual. Interestingly, even under “stable” flow waveform conditions, the LCX in the AOCA patient showed a larger flow rate, further increasing the burden on the anomalous vessel. Similarly, as shown in Figure 5A, when the LAD of the AOCA patient exhibited very mild stenosis, an obvious increase in the TAWSS was observed in the proximal part of the anomalous LCX, even beyond the plaque site, which may lead to potential adverse events. Therefore, the disrupted LAD–LCX communication structure in the AOCA patient may weaken the coronary system’s adaptive regulation capacity. In fact, Corrado et al. documented a case of sudden death in a patient with an LCX originating from the right sinus, where, despite no evidence of obstructive stenosis, the patient still exhibited symptoms of myocardial infarction and ultimately died [5]. Our results might explain why even very mild stenosis can lead to adverse cardiovascular events.

This study has several limitations. First, due to the rare clinical incidence of AOCA, this study only included one case of an AOCA patient to conduct hemodynamics analysis, and our findings still require future validation with a larger patient cohort. Second, this study employed the rigid wall assumption, as it maintains reasonable accuracy for time-averaged parameters, such as TAWSS and OSI, as well as flow patterns, while significantly reducing computational time compared to fluid–structure interaction (FSI) simulations. Although this assumption may influence the absolute values of hemodynamic parameters, its uniform application across all cases in this study ensures that differences in hemodynamic parameters between models primarily arise from the abnormal coronary morphology of the AOCA patient, making these trends representative. Future studies could consider applying FSI to enhance the accuracy of the results. Third, while the LPN model effectively captures the boundary conditions of AOCA patients’ coronary arteries and simulates their physiological behavior, incorporating real patient data from 4D Flow MRI could enhance accuracy.

## 5. Conclusions

Coronary arteries in the AOCA patient exhibited characteristics of high pressure and high shear stress, and stenosis further exacerbated the burden on the abnormal artery, increasing the risk of atherosclerosis. Clinicians should pay extra attention to AOCA patients, particularly those with stenosis, by enhancing follow-up protocols and implementing preventive measures to mitigate the likelihood of adverse cardiovascular events.

## Figures and Tables

**Figure 1 bioengineering-11-01196-f001:**
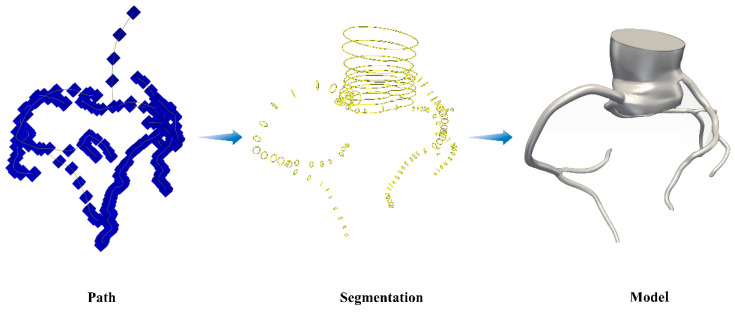
Procedures of the 3D model reconstruction based on patient-specific CTA imaging.

**Figure 2 bioengineering-11-01196-f002:**
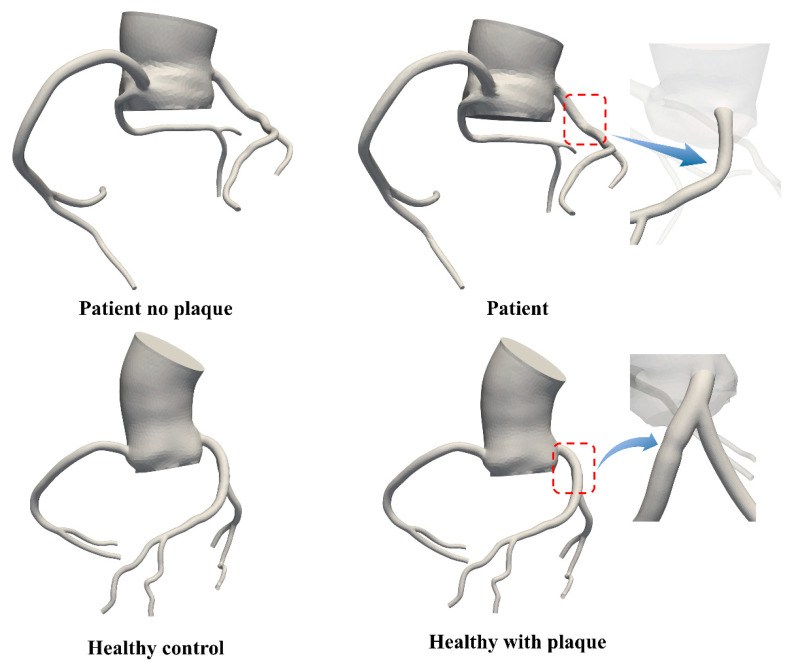
Schematic representation of the models included in this study.

**Figure 3 bioengineering-11-01196-f003:**
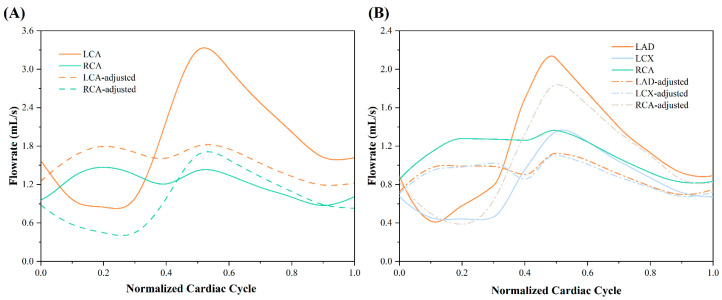
The schematic diagram of coronary waveform adjustments for the healthy and patient models. (**A**) healthy model; (**B**) patient model.

**Figure 4 bioengineering-11-01196-f004:**
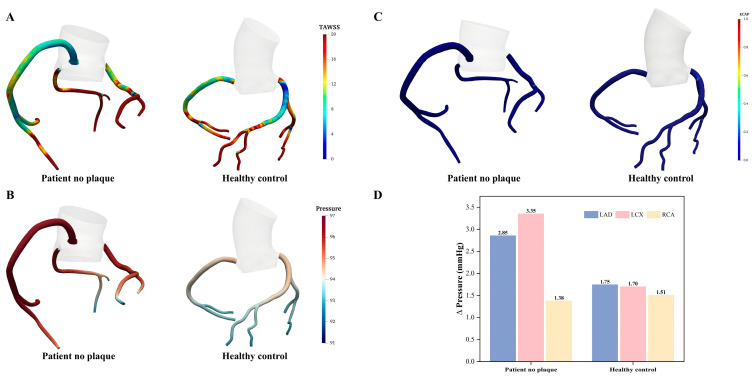
Hemodynamic differences between the patient and healthy control models without stenosis. (**A**) TAWSS (in dyne/cm^2^); (**B**) Time-averaged wall pressure (in mmHg); (**C**) ECAP distributions; and (**D**) Average pressure drop across the coronary inlet and outlets.

**Figure 5 bioengineering-11-01196-f005:**
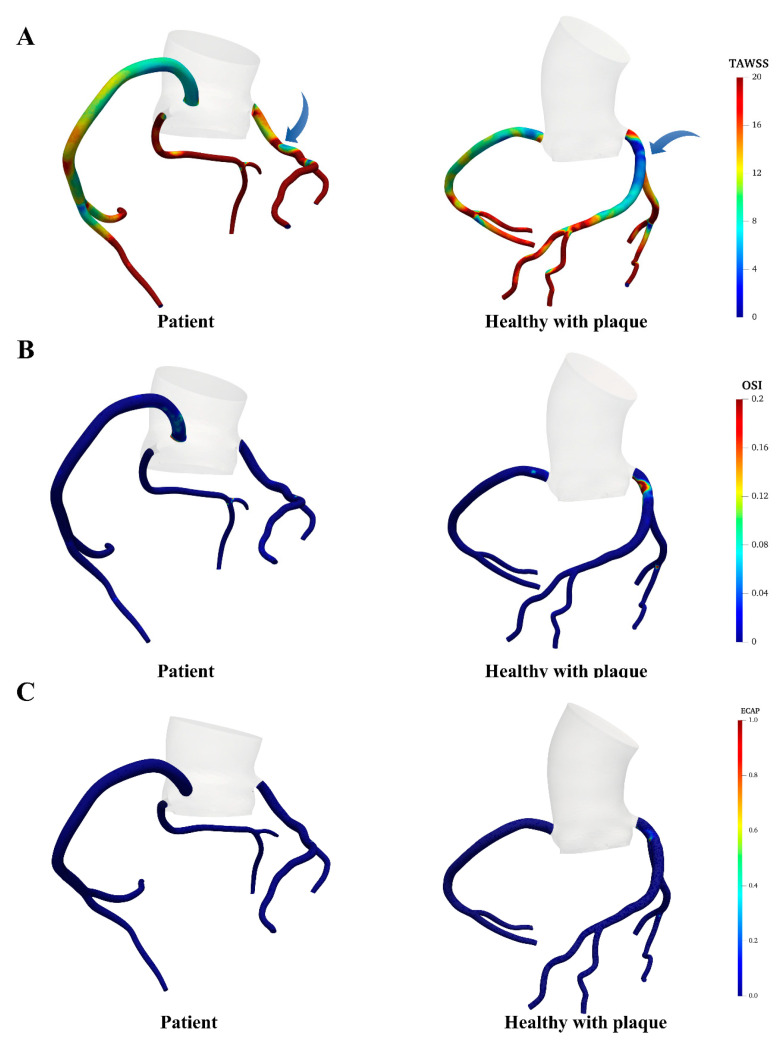
Contour maps of the distribution of (**A**) TAWSS; (**B**) OSI; and (**C**) ECAP between the AOCA patient and healthy control.

**Figure 6 bioengineering-11-01196-f006:**
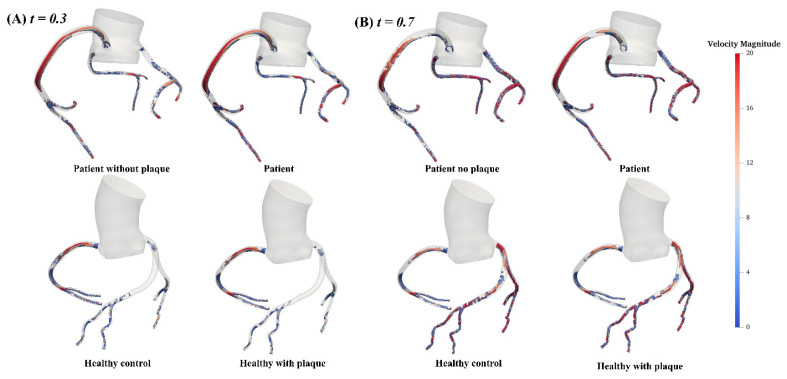
The iso-contour map of vortex Q-criterion distribution during the coronary blood flow acceleration phase (**A**) and deceleration phase (**B**) for the AOCA patient and healthy individual.

**Figure 7 bioengineering-11-01196-f007:**
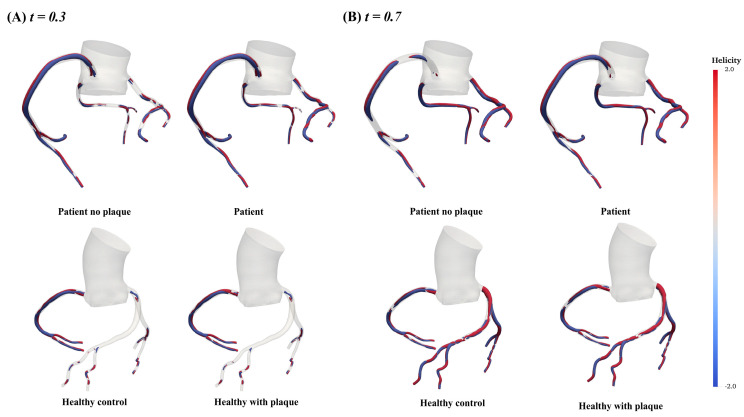
Iso-contour maps of helical flow distribution in the LAD, LCX, and RCA during the coronary flow acceleration and deceleration phases for the AOCA patient and healthy model.

**Figure 8 bioengineering-11-01196-f008:**
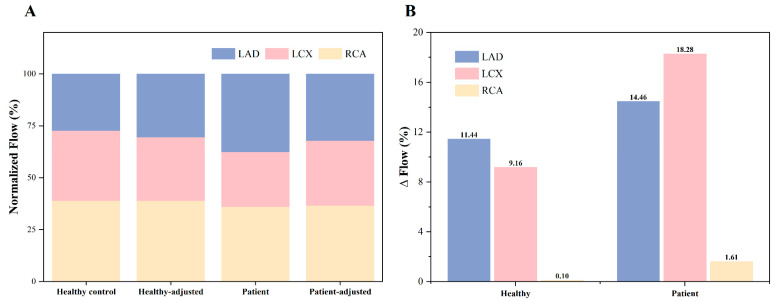
Flow distribution and changes in the patient and normal coronary models before and after waveform adjustment. (**A**) Flow distribution; (**B**) Flow changes in the LAD, LCX, and RCA after adjustment in coronary waveform.

**Table 1 bioengineering-11-01196-t001:** Baseline characteristics of the AOCA patient and healthy individual included in this study.

		AOCA Patient	Healthy Control
**Age, years**		73	53
**Gender**		male	female
**BMI**		23.1	20.3
**BP, mmHg**	Systolic	122	
	Diastolic	75	

AOCA, anomalous origin of coronary artery; BMI, body mass index; BP, blood pressure.

**Table 2 bioengineering-11-01196-t002:** Mesh independence analysis of the Patient model.

Size	Number of Elements	Q_LAD_	∆P
0.06	642,253	1.22	3.87
0.04	1,553,994	1.14	3.44
0.03	2,988,571	1.12	3.45

The mesh size is measured in millimeters (mm), the flow rate in milliliters per second (mL/s), and the pressure drop in millimeters of mercury (mmHg). LAD, left anterior descending aorta.

## Data Availability

The original contributions presented in the study are included in the article. Further inquiries can be directed to the corresponding authors.

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
