# Peer review of "The Association Between the Hemodynamics in Anomalous Origins of Coronary Arteries and Atherosclerosis: A Preliminary Case Study Based on Computational Fluid Dynamics"

_bioengineering, 2024, doi:10.3390/bioengineering11121196_

Round 1
Reviewer 1 Report
Comments and Suggestions for Authors
The review of manuscript " Association Between Anomalous Origin of Coronary Arteries and Atherosclerosis: A Preliminary Hemodynamic Study Based on CFD " by Wei et al.
They have looked at whether the presence of strenosis in AOCA patients leads to a risk of atherosclerosis using the CFD model based on open source Sim Vascular. The authors have explained the work for clinicians well but there are certain details which would be needed for general audience.
1. what the settings used in the Sim Vascular while reconstruction of CTA scans into obj files or stl files? How are the surfaces of arteries made smooth?
2. why were the vessel walls be modelled as rigid as in reality the vessel walls are flexible?
3. To make the work complete, I am of the opinion that authors should explain LPN model and other inputs for boundary conditions in complete in the present manuscript itself.
Reviewer 2 Report
Comments and Suggestions for Authors
Overall, it is an interesting and well-written article. It seeking to better understand a gap in the literature, however, authors do not make it clear that it is a case study, often confusing the reader. Furthermore, some aspects of the discussion should be viewed with caution given the study design. In this way results generalization is limited.
Reding the text is not clear how many individuals were evaluated. This is a fundamental information to study comprehension.
Title: add information about study design
Abstract
Change subjects by individuals
Add study design
Generalization is limited by study design.
Introdutivo
Add in objective and hypothesis more information about which individuals will be analyzed.
Methods
Add information about study design
Are there any inclusion criteria related with age or myocardial ischemia history?
Add information about exclusion criteria.
Results
Add more information about participants, as age, sex, physical activity practice, weight. Are they similar and comparable?
Make it clear in results that only one AOCA patient and one healthy individual were evaluated.
Was any statistical analysis performed to confirm a significant difference? Based in which parameters the authors confirm de significant difference?
Discussion
In the sentence “On the other hand, a greater pressure drop across the inlet and outlet suggests that the heart has to work harder to maintain physiological demands, increasing the myocardial load and placing additional stress on the vessels, which raises the risk of atherosclerosis and creates a vicious cycle” add references.
Add information about results clinical relevance.
“Our results reveal large areas of high TAWSS in the LAD and LCX of AOCA patients” just ne patient with AOCA was evaluated.
“Consequently, the intraluminal hemodynamic environment in AOCA patients makes them more susceptible to atherosclerosis compared to individuals with normally originated coronary arteries” authors cannot confirm this sentence once only one AOCA patient and one healthy individual were evaluated.
What the authors mean with “individuals with normally originated coronary arteries”?
“Moreover, the impact of flow waveform on coronary perfusion cannot be overlooked, and the extent of this impact is related to the geometry of the vessel. Under normal conditions, the collateral communication between the LAD and LCX helps regulate blood flow distribution and maintain the myocardial perfusion demand” Add references.
The most important problem in the study is generalization usually made.
Round 2
Reviewer 2 Report
Comments and Suggestions for Authors
I really appreciated the authors' word to change the manuscript and improve it quality. However, some points sill not described, sometimes still not clear the study design that can make the readers confuse. Come point were cited by the authors as changed but I can not identify changes un manuscript. On the other hand, statistical analysis is not adequate. One just a case report authors cannot talk about statistical significance or perform any statistical test; it is there is little data to carry out any statistical test.
Due to these problems that may confuse the reader and regarding conclusions about the study, I believe that this study should not be published.
